# Circulating Neoplastic-Immune Hybrid Cells Are Biomarkers of Occult Metastasis and Treatment Response in Pancreatic Cancer

**DOI:** 10.3390/cancers16213650

**Published:** 2024-10-29

**Authors:** Ranish K. Patel, Michael Parappilly, Hannah C. Farley, Emile J. Latour, Lei G. Wang, Ashvin M. Nair, Ethan S. Lu, Zachary Sims, Byung Park, Katherine Nelson, Skye C. Mayo, Gordon B. Mills, Brett C. Sheppard, Young Hwan Chang, Summer L. Gibbs, Adel Kardosh, Charles D. Lopez, Melissa H. Wong

**Affiliations:** 1Department of Surgery, Division of Surgical Oncology, Oregon Health & Science University (OHSU), Portland, OR 97239, USA; patera@ohsu.edu (R.K.P.);; 2Department of Cell, Developmental and Cancer Biology, OHSU, Portland, OR 97201, USA; 3Biostatistics Shared Resource, Knight Cancer Institute, OHSU, Portland, OR 97239, USA; 4Department of Biomedical Engineering, OHSU, Portland, OR 97201, USA; 5Knight Cancer Institute, OHSU, Portland, OR 97201, USA; 6Gastrointestinal Clinical Trials, Knight Cancer Institute, OHSU, Portland, OR 97239, USA; 7Division of Oncological Sciences, Knight Cancer Institute, OHSU, Portland, OR 97239, USA; 8Department of Surgery, Division of General Surgery, OHSU, Portland, OR 97239, USA; 9Department of Medicine, Division of Medical Oncology, OHSU, Portland, OR 97239, USA

**Keywords:** circulating hybrid cells, pancreatic cancer, cancer biomarker, early detection, liquid biopsy

## Abstract

Pancreatic ductal adenocarcinoma (PDAC) is an aggressive cancer that is challenging to diagnose and treat. Current methods to measure disease burden, monitor treatment response, and predict disease progression have significant limitations. This study explores the utility of circulating neoplastic-immune hybrid cells (CHCs) as a novel blood-based biomarker. We find that pre-operative CHC levels can predict the presence of metastatic disease that may only be detected during surgical exploration, as well as identify patients who are likely to experience rapid metastatic progression after surgery. Further, CHC levels and protein expression patterns are reflective of disease response to chemotherapy. Collectively, these results suggest that CHCs could serve as crucial indicators of disease status and treatment efficacy, offering great potential to enhance survival outcomes for patients with PDAC.

## 1. Introduction

Pancreatic ductal adenocarcinoma (PDAC) is one of the most lethal malignancies worldwide and is predicted to become the second leading cause of cancer-related deaths in the United States by 2030 [1,2,3]. Most patients are diagnosed with advanced, late-stage disease due in part to the lack of early detection tools, aggressive biology, and indolent clinical presentation. Furthermore, despite advances in surgical technique, neoadjuvant and adjuvant treatment strategies, and multidrug and targeted systemic therapies, survival has remained dismal, with an estimated 5-year overall survival of 11% [1].

Several important challenges exist in the management of patients with PDAC. While surgical resection represents the only presumptive curative treatment modality, patients are frequently under-staged by standard cross-sectional imaging (computed tomography (CT) and magnetic resonance (MR) imaging), and 10–20% of patients may be found to have occult metastatic disease upon operative exploration [4,5,6,7]. Further, disease recurrence following margin-negative surgical resection is common, with as many as 20–40% of patients rapidly developing detectable disease within 6 months after surgery, which may represent synchronous occult metastatic disease unidentified at the time of operation [8,9,10,11,12,13]. With improved disease characterization, these patients could be spared morbid and futile operative interventions and receive more appropriate first-line systemic therapies. Furthermore, given the increased adoption of neoadjuvant and adjuvant therapies for patients with PDAC, a lack of reliable response and surveillance measures presents another major obstacle. Radiographic imaging is generally offered at 3- to 6-month intervals and heavily relies on size measurements as imperfect surrogates of tumor biology [14,15]. Carbohydrate antigen 19-9 (CA 19-9) remains the gold standard blood-based biomarker, though it is limited in specificity as a measure of disease response and relapse as CA 19-9 levels are influenced by potentially confounding benign clinical conditions (biliary obstruction, pancreatitis, cirrhosis) and is not produced at baseline in approximately 10% of the general population [16,17,18]. Thus, there is a critical need to develop biomarkers that can better stage disease, predict rapid recurrence, and more reliably measure therapeutic effects.

Circulating neoplastic-immune hybrid cells (CHCs) are a unique population of disseminated tumor-derived cells that functionally express both neoplastic and immune cell attributes and are detected in the peripheral blood of cancer patients. CHCs harbor co-expression of tumor and leukocyte proteins [15,19,20,21,22,23,24,25,26], and thus they are distinct from conventional circulating tumor cells (CTCs), which are devoid of immune cell identity. Multiple mechanisms for hybrid cell formation are proposed [22,27,28,29,30], including cellular fusion [20,31,32]. Cell fusion-derived hybrid cells that disseminate into circulation retain proteomic and genomic attributes of the primary tumor, making them a promising biomarker of the tumor burden and status [19,20,33].

CHCs have promising translational value as blood-based cancer biomarkers. CHCs are uniquely detected in cancer patients (including those with PDAC) when compared to healthy controls [19,20,21,34], with elevated numbers correlating with increasing tumor burden and poor survival [21]. Furthermore, in patients with PDAC, isolated CHCs have been shown to harbor KRAS mutations consistent with the primary tumor from which they derived [19,20]. Interestingly, these relationships have not been reliably established or are difficult to evaluate in CTCs, owing to their relative rarity. For example, as few as 1 CTC may be detected from a 7.5 mL peripheral blood sample obtained from patients with high tumor burden and frequently may not be detected at all in those with early-stage disease [35,36,37,38,39]. In contrast, CHCs are readily detectable at levels often an order of magnitude greater than CTCs in patients with early stage to metastatic disease [19,20,21].

In this study, we evaluate the utility of CHCs as a quantitative and qualitative blood-based biomarker in patients undergoing multi-modality treatment of PDAC. We demonstrate that CHCs provide real-time insight into disease burden, recurrence risk, and treatment response.

## 2. Materials and Methods

### 2.1. Human Specimens

Human specimens were collected and analyzed under approved protocols (OHSU Biolibrary and the Oregon Pancreatic Tumor Registry) under the ethical requirements and regulations of the institutional review board at Oregon Health & Science University. Peripheral blood samples were obtained from patients with PDAC immediately prior to scheduled treatment or procedure from these groups: patients undergoing surgical resection, patients enrolled in the NeoOPTIMIZE phase II clinical trial (NCT04539808), and those enrolled in the window of opportunity for metastatic PDAC (WOO-M) phase I clinical trial (NCT04005690). The NeoOPTIMIZE clinical trial investigates the efficacy of adaptive neoadjuvant therapeutic regimens for those with localized PDAC. Longitudinal peripheral blood samples were collected at multiple pre-established time points within the trial protocol. The WOO-M clinical trial studies pre- and post-treatment core needle biopsies of metastatic PDAC lesions to the liver to facilitate biomarker discovery. Patients were treated with 10 days of either a polo-like kinase 1 (PLK-1) inhibitor or a mitogen-activated protein kinase (MEK) inhibitor. For patients on the WOO-M trial, a peripheral blood sample was obtained immediately prior to a pre-treatment biopsy and at the time of the post-treatment biopsy. Tissues collected from pre- and post-treatment biopsies were matched to collected peripheral blood samples. No patients were cross-enrolled to both the NeoOPTIMIZE and WOO-M clinical trials. Blood samples from self-reported healthy volunteers were analyzed as controls.

### 2.2. Peripheral Blood Sample and Tumor Tissue Preparation

Peripheral blood: As previously described [34], 10–20 mL of peripheral blood was collected into heparinized vacutainer tubes (BD Biosciences, Franklin Lakes, NJ, USA) and processed within 4 h of collection. Standard Ficoll–Paque density centrifugation (GE Healthcare, Chicago, IL, USA) was used to isolate peripheral blood mononuclear cells (PBMCs). The cells were then adhered to poly-D-lysine-coated slides (Fisher Scientific, Hampton, NH, USA) through incubation at 37 °C for 15 min, permeabilized with Triton-X, and then fixed with 4% paraformaldehyde.

Tumor tissue: Tumor tissue was prepared using established protocols [33]. Formalin-fixed paraffin-embedded metastatic PDAC tissue sections (5 µm) were deparaffinized with xylene baths and then rehydrated in sequential graded ethanol baths. Tissue slides were incubated in citrate buffer (Sigma-Aldrich, St. Louis, MO, USA, pH 6) for 30 min at 100 °C, Tris–HCl buffer (Invitrogen, Waltham, MA, USA, pH 8) at 100 °C for 10 min, cooled to room temperature, and then washed in phosphate buffered saline (PBS) at room temperature to complete antigen retrieval.

### 2.3. Immunohistochemical Staining for CHC Enumeration

Per established protocol [34], PBMC slides were blocked with 5% bovine serum albumin (BSA) blocking buffer (2.5 M CaCl_2_, 1% Triton-X-100, 1% bovine serum albumin in PBS) for 30 min, stained with fluorescent-conjugated antibodies for pan-cytokeratin (CK, AE1/AE3, Invitrogen, Waltham, MA, USA), and CD45 (HI30, Life Technologies, Carlsbad, CA, USA), and then counterstained with the nuclear dye, 4,6-diamidino-2-phenylindole (DAPI). On each slide, one region was isolated using a hydrophobic barrier marker and left unstained to be utilized as a reference control for detecting autofluorescence during analysis.

### 2.4. Image Capture and Data Processing for CHC Enumeration

Slides were digitally imaged using a Zeiss AxioScan. Z1 light microscope (Oberkochen, Germany). CHCs were defined by co-expression of CK and CD45 (CK^+^/CD45^+^); CTCs were defined by CK expression only (CK^+^/CD45^−^). Fluorescence histogram thresholds were defined using the unstained regions of each patient slide. The Zeiss Efficient Navigation (ZEN) blue software version 3.5 (Carl Zeiss AG, Oberkochen, Germany) was used to enumerate CHCs and CTCs with a semi-automated approach, blinded to the clinical status of the specimen. For each peripheral blood sample, at least 50,000 PBMCs were evaluated, and CHC counts were standardly reported normalized to a concentration per 50,000 total PBMCs. For high-resolution imaging, patient PBMCs were imaged utilizing confocal microscopy with a laser scanning microscope (Zeiss LSM 880) equipped with an Airyscan array detector and automatic Airyscan processing ZEN Black software version ZEN 2.3 SP1 (Zeiss). Standard laser lines (488 and 561 nm) were used to excite AF488 and AF570, respectively. Fast Airyscan images were captured with a 63× oil objective and analyzed using ImageJ/FIJI (NIH).

### 2.5. Cyclic Immunofluorescence (CyCIF) Analysis of Blood and Tissue Specimens

Antibody conjugation to oligonucleotides (Ab-oligo) was performed as previously described, which allowed for multiplex phenotyping of a single patient sample [40]. Briefly, the SiteClick™ Antibody Azido Modification kit (ThermoFisher, Waltham, MA, USA) was used to conjugate a unique 28mer oligonucleotide single-stranded (ssDNA) docking strand to the Fc region of each antibody. For the detection of the targeted marker, 26-base complementary ssDNA imaging strands (IS) were utilized, each strand labeled with fluorophores and photo-cleavable linkers at the 5′ and 3′ termini. All oligonucleotides were purchased from IDT (Coralville, IA, USA). Refer to Appendix A for antibody and oligonucleotide information.

CyCIF staining was completed as previously described [40]. Samples were first counterstained with DAPI and imaged to establish background autofluorescence levels. Ab-oligos were then applied to samples in a single staining step. Excess unbound Ab-oligo was washed off in three separate 2× saline sodium citrate (SSC) solution baths for five minutes each. Four imaging strands (IS) were added to the specimens, each bound with spectrally distinct fluorophores DAPI (Zeiss 96 HE), Alexa Fluor 488 (AF488, Zeiss 38 HE), AF555 (Zeiss 43 HE), AF647 (Zeiss 50), and AF750 (Chroma 49007 ET Cy7), and then slides were imaged. Subsequently, the fluorescent signal was photocleaved using ultraviolet (UV) light treatment for 20 min using a UV light box. The specimens were rewashed with 2× SSC before a subsequent round of imaging to confirm the complete removal of the fluorescence signal. Tissues and PBMCs were then stained with a second round of distinct conjugated fluorescent antibodies as previously described. Conjugated fluorescent antibodies were stained in the second round after Ab-oligos, as these standard antibodies were conjugated to fluorophores, which were not photocleavable and could not be removed to allow for sequential rounds of staining.

### 2.6. Image Capture and Data Processing for CyCIF Blood and Tissue Specimens

PBMCs and tissues were imaged utilizing a Zeiss AxioScanner. Z1 light microscope. Images obtained from successive rounds of cyclic staining were registered using QiTissue software v.1.4.0 (Quantitative Imaging Systems, Pittsburgh, PA, USA) or a feature-based image registration method. For the feature-based image registration, each cycle’s images were individually registered to the first cycle using an affine transformation, calculated by identifying matching key points between the DAPI channels of each cycle [41]. Key point matches were determined using the OpenCV [42] implementation of scale-invariant feature transform (SIFT) feature extraction [43]. Images of direct immunofluorescence, Ab-oligo plus IS, and Ab-oligo plus IS after UV cleavage were then segmented in QiTissue with DAPI as the nuclear marker. Regions of interest (ROIs) with representative antibody staining were randomly selected to calculate the signal-to-background ratio and for staining validation. Control samples, specifically those with IS only and UV-cleaved, served as a baseline for defining the threshold of positive antibody staining. Positive expression was determined as an average whole-cell intensity value exceeding the positive staining threshold, which was obtained by normalization to the IS only or UV-cleaved controls. Cells that exhibited suboptimal segmentation or negative cells with segmentation intersecting that of a positive cell’s expression were omitted from the analysis. Tissue section peripheries and PBMC-well boundaries were omitted from the analysis, along with regions exhibiting imaging artifacts, such as out-of-focus areas and bubbles. Within QiTissue, cells displaying pronounced autofluorescence were excluded based on their average whole-cell fluorescence derived from the background channel in round 0. Hybrid cells were identified by the co-expression of CD45 and CK and then phenotyped by expression of phosphorylated protein kinase B (pAKT) and epidermal growth factor receptor (EGFR). In patients receiving therapy with a MEK inhibitor, stimulator of interferon response cGAMP interactor (STING) expression was also analyzed.

### 2.7. Definitions

All information regarding laboratory data, radiographic data, pathologic data, clinical history, and treatment schedule was extracted directly from the electronic health record. Clinical and pathologic staging were reported utilizing the American Joint Committee on Cancer staging criteria for exocrine pancreatic cancer. Rapid recurrence of disease was defined as metastatic recurrence within 6 months of microscopically negative (R0) margin pancreatectomy. Assessment of pathologic tumor response to neoadjuvant therapy was standardly reported in patient pathology reports utilizing the modified Ryan scheme for tumor regression score (TRS): TRS 0 (complete response), TRS 1 (near-complete response), TRS 2 (partial response), and TRS 3 (poor or no response) [44,45]. All tissue samples retrieved from core needle liver biopsies of metastatic PDAC lesions were standardly reviewed for Ki67 expression by a single expert pathologist and extracted from pathology reports as percentage positive of total tumor cells.

### 2.8. Statistical Analysis

Descriptive statistics (median and inter-quartile range for continuous variables; frequencies and percentages for categorical variables) were used to describe demographic and clinical characteristics. Neoadjuvant groups were compared using Fisher’s exact test and the Wilcoxon–Mann–Whitney test. Comparisons of continuous data between more than two groups were completed using the Kruskal–Wallis test. If a significant result was found using the Kruskal–Wallis test, the Dwass–Steel–Critchlow–Fligner test procedure was used to make multiple pairwise comparisons while providing family-wise error rate protection. The specific comparison of mean preoperative CHCs vs. tumor response groups was conducted utilizing a permutation test with 10,000 permutations, given the small sample size and the less strict assumptions of this test. Significance was set at *p*-value < 0.05. Analysis was performed using R: A Language and Environment for Statistical Computing [46]. For longitudinal analyses, CHC levels were organized chronologically per patient and correlated to patient clinical status and tumor pathology. For CHC and paired tumor tissue phenotypic analysis, descriptive statistics were utilized given limited sample size and statistical fragility.

## 3. Results

### 3.1. Preoperative CHC Levels Are Predictive of Radiographic Occult Metastases and Rapid Metastatic Recurrence Following Pancreatectomy

To evaluate the utility of CHCs as predictors of unrecognized metastatic disease, peripheral blood samples from 42 patients with PDAC were collected immediately prior to scheduled pancreatectomy (Table 1). The median age of the cohort was 67 years; 18 (42.9%) were female, and 35 (83.3%) were of white and non-Hispanic race and ethnicity, respectively. Fourteen (33.3%) patients were not treated with neoadjuvant therapy (NAT), twenty-four (57.1%) were treated with neoadjuvant chemotherapy only, and four (9.5%) were treated with neoadjuvant chemoradiation. CHCs were reliably found across the patient cohort by their co-expression of CK^+^ and CD45^+^ as detected by standard fluorescence and confocal microscopy (Figure 1). A median of 15.49 CHCs (IQR 7.11–34.64) per 50,000 PBMCs, and at least one CHC was identified in 93% of the study cohort. Preoperative CHC numbers and CA 19-9 values were similar in patients treated with NAT compared to those not treated with NAT (*p* = 0.393 and *p* = 0.408, respectively). Similar to our previously published data, disseminated CHC counts increased with tumor burden, and patients with localized (lymph node negative; stages IA, IB, and IIA), locoregional (lymph node positive; stages IIB and III), and occult metastatic (stage IV) disease discovered upon surgical exploration harbored elevated numbers of CHCs relative to healthy subjects (n = 17). Preoperative CA 19-9 levels did not statistically differ by tumor burden across our cohort (Figure 2).

Given that CHC numbers increase with tumor burden in patients with PDAC, we evaluated whether CHC levels could accurately identify and discriminate patients with occult metastatic disease, which was not identified through a robust, multidisciplinary preoperative clinical evaluation. Of the 42 patients included in the study cohort, 8 (19%) were found to have detectable metastatic disease at the time of surgery, with peritoneal carcinomatosis being the most common site (62.5%) of distant disease, and thus pancreatectomy was aborted (Table 1). When comparing those with surgically detectable metastatic disease to those without, the metastatic group had a mean CHC count of 65 (IQR 34.6–57.9) as compared to 22.8 (IQR 6.8–23.4) in patients without detectable metastatic disease (*p* < 0.001). Conversely, preoperative CA 19-9 levels did not significantly differ between the two groups, with mean 159.10 U/mL and 190.81 U/mL (*p* = 0.162) when comparing the metastatic and non-metastatic groups, respectively (Figure 3A,B).

Furthermore, given that preoperative CHC numbers correlated with tumor burden and were significantly higher in patients with metastatic disease detected at surgery, we explored the possibility that preoperative CHC counts could also identify patients who experience rapid metastatic PDAC recurrence with the rationale that these patients may have harbored clinically and surgically unidentifiable metastatic disease at the time of surgical resection. Within the initial study cohort, 30 patients did not have surgically identified occult metastases and had at least 6 months of postoperative follow-up. Of these patients, six patients (20%) experienced rapid metastatic recurrence to the liver, lungs, and/or peritoneum. Patients who experienced rapid recurrence had significantly higher preoperative CHC numbers when compared to patients who did not have recurrent disease, with 74.4 CHCs (IQR 38.9–91.3) vs. 11.52 CHCs (IQR 5.8–16) (*p* < 0.001), respectively. Interestingly, preoperative CHC numbers were at similar levels in patients with rapid recurrence and occult metastases (*p* = 0.964). There was no statistically significant difference (*p* = 0.461) in preoperative CA 19-9 levels in the three groups (i.e., without occult metastases, with occult metastases, with rapid recurrence) (Figure 3C,D).

### 3.2. Pathologic Response to Neoadjuvant Therapies Correlates with Chc Levels

To evaluate how CHC numbers change in response to NAT in patients without occult or rapidly recurrent disease, we evaluated preoperative CHC numbers (i.e., specimens obtained on the day of surgery, following completion of NAT), and compared these numbers to the pathologic response as assessed by standardized tumor regression scores (TRS). Of the 42 patients in the cohort, 17 patients were treated with NAT, underwent successful pancreatectomy (and thus did not have occult metastases), did not experience rapid recurrence, and had pathologic TRS scores reported. Within the cohort, 1 patient had a complete response (TRS 0), 3 patients had a near-complete response (TRS 1), 12 patients had a partial response (TRS 2), and 1 patient had poor or no response (TRS 3). Those with TRS of 0 or 1 had significantly lower CHC counts when compared to those with TRS of 2 or 3, with 1.7 vs. 13.06 CHCs, respectively (*p* = 0.008).

Given that CHC counts were significantly lower in patients with greater responses to NAT, we evaluated the utility of CHCs as a real-time readout of NAT response. In collaboration with the investigators of the NeoOPTIMIZE clinical trial (NCT04539808), which studies the efficacy of adaptive NAT regimens for those with localized PDAC, we prospectively and longitudinally evaluated CHC numbers in eight patients and temporally correlated CHC numbers to treatment with NAT and ultimate pathologic response following pancreatectomy (Figure 4). Patients were divided into three groups: those with partial pathologic response (TRS 2), those with no or poor response (TRS 3), and those who developed distant metastatic disease while receiving NAT; no patients evaluated ultimately had complete or near-complete responses to therapy (TRS 0 or 1).

Four patients demonstrated a partial response to NAT (TRS 2). In all four patients, CHC numbers decreased following the initiation of systemic chemotherapy and ultimately reached undetectable levels. Subsequently, in all four patients, CHC counts increased in the interval between completion of systemic chemotherapy and surgical resection. Three patients demonstrated poor or no pathologic response to treatment (TRS 3). In these patients, despite the initiation of systemic chemotherapy, CHC levels remained detectable throughout treatment, as numbers initially decreased in two patients but ultimately trended upwards in all patients by the time of surgical resection. In the single patient who developed distant metastatic disease while receiving NAT, CHC levels initially decreased to zero after initiation of systemic therapy, though subsequently increased prior to discovery of metastases, while still receiving systemic chemotherapy.

Most patients in the TRS 2 and 3 groups saw a downtrend in CA 19-9 levels with NAT, suggesting a treatment response. However, multiple patients paradoxically either experienced initial increases in CA 19-9 after initiation of therapy, no clear trend in CA 19-9 levels throughout treatment, or were CA 19-9 non-formers. In the patient who developed metastatic disease on therapy, CA 19-9 levels increased and then decreased following initiation of systemic chemotherapy, though ultimately trended upwards prior to the development of distant metastatic disease. Notably, two patients experienced marked increases in CHC levels at the time of development of cholecystitis and a liver abscess, respectively, suggesting the possibility that CK^+^/CD45^+^ CHC levels may additionally be affected by severe, non-malignant inflammatory insults. However, in the case of the patient who developed cholecystitis, CHC levels decreased following treatment with cholecystectomy but remained elevated relative to baseline, perhaps reflecting the underlying primary tumor resistance to NAT (Figure 4).

### 3.3. Disseminated CHCs Reflect Phenotypic Changes Also Detected in the Metastatic Tumor in Response to Short-Interval Systemic Therapies

To determine if disseminated CHCs could be used to monitor tumor-specific biological responses to systemic therapy, we employed multiplexed CyCIF on paired metastatic PDAC tumor and peripheral blood specimens at two different time points. Leveraging patient samples from those enrolled in the WOO-M trial (NCT04005690), which aims to study biological activity of metastatic PDAC lesions to the liver in response to targeted therapies, metastatic tumor tissue, and peripheral blood specimens, were collected at baseline and after 10 days of treatment with systemic targeted therapy with either a PLK-1 inhibitor (n = 2) or MEK inhibitor (n = 2), per the trial protocol.

To demonstrate whether CHC protein expression reflects inherent metastatic tumor tissue proliferative response to systemic therapy, we evaluated Ki67 expression in paired samples. After completion of 10 days of trial treatment, the proportion of CHCs with positive Ki67 expression increased in one patient, decreased in two patients, and remained unchanged in the fourth patient. These observed changes in the proliferative index in CHCs were concordant with the formal pathologic evaluation of proliferation in the corresponding tumor tissue samples (Figure 5A,B).

In addition, to evaluate if protein expression that correlates with treatment response remains consistent between CHCs and tissue biopsies and therefore provides a real-time readout, we compared expression of relevant PDAC-specific pathway proteins. This included the STING pathway (stimulator of interferon response cGAMP interactor), epidermal growth factor receptor (EGFR), and phosphorylated protein kinase B (pAKT). Upregulation of STING relates to improved type I IFN-dependent tumor cytotoxicity, and MEK inhibition enhances STING agonism and downstream cell death and tumor regression [47,48]. Overexpression of EGFR is associated with increased tumor growth and invasiveness [49,50]. Constitutive activation of pAKT is frequently observed in PDAC, correlating with poor prognosis, chemoresistance, and enhanced tumor survival [51,52].

Of the two patients who were treated with a MEK inhibitor, one tumor had an increase in STING expression following treatment, while STING expression increased in the disseminated CHC populations in both patients. EGFR expression was concordant between CHCs and tumor tissue in three of four patients, all of whom saw decreases in response to treatment. When evaluating pAKT expression, similarly, CHC and tumor tissue changes mirrored one another in three of four patients (Figure 5C,D).

Overall, these findings indicate that disseminated CHC phenotypes reflect the changes that occur within the intact tumor and can be influenced by short-course systemic therapies. Thus, this data indicates that CHC evaluation could be leveraged to assess the efficacy of systemic therapies.

## 4. Discussion

Despite significant progress in the development of therapeutic and diagnostic strategies over the past two decades, patients with PDAC continue to experience dismal prognoses. The lack of survival advances relates in part to an inability to identify occult disease early and to monitor changes in disease status throughout treatment. Current biomarkers such as CA 19-9 and cross-sectional imaging techniques have inherent limitations; thus, there is a pressing need for additional readily accessible biomarkers that can synergize with current practice to enhance informed care. CHCs are disseminated neoplastic-derived cell populations, with promise to be developed as biomarkers that can overcome many of these challenges. In the present study, we demonstrate that CHCs have great potential as a blood-based analyte, as they are predictive of occult metastatic disease and provide invaluable information regarding tumor response to systemic therapies.

In our study cohort consisting of patients who were treated with and without NAT, preoperative CHC levels successfully identified those found to have occult metastatic disease at surgical exploration and rapid recurrence within 6 months of curative-intent surgical resection. CHC levels were similar between these two groups; given that the phenomena of rapid recurrence may represent oligometastatic disease extant at the time of pancreatectomy, it is notable that these two groups appear to be similar in preoperative disseminated CHC numbers. Therefore, CHCs may represent a novel biomarker to detect patients with radiographically and clinically occult micrometastatic disease earlier than conventional staging measures and have promise to be leveraged to tailor therapeutic strategies. Importantly, neither occult metastatic disease nor rapid recurrence were reliably differentiated when comparing preoperative CA 19-9 levels in our cohort. This finding is not surprising given that CA 19-9 levels can vary greatly depending on tumor biology, confounding conditions, and baseline patient characteristics, in the context of our sample size. Larger series have demonstrated relationships between CA 19-9 levels with occult metastatic disease and, to a lesser degree, rapid recurrence, though cut-off thresholds and reported sensitivity/specificities differ and are not well established for either use case [5,8,36,53,54,55,56]. Conventional CTCs have demonstrated promise as potential biomarkers predictive of occult metastasis; in a study of 100 patients with PDAC, CTCs (though not CA 19-9 levels) correlated with disease stage and successfully predicted those with occult metastatic disease when compared to those without, regardless of neoadjuvant treatment status. However, at least one CTC was found in 78% of their patient cohort and was skewed towards later-stage patients [36]. This contrasts with 93% of patients in our cohort who were found to have at least one CHC, with the remaining 7% lacking CHCs in the setting of complete or near-complete responses to NAT. Circulating tumor DNA (ctDNA) has similarly demonstrated efficacy in predicting occult metastatic disease; in a study of 142 patients, ctDNA was found to be detectable in 41% of those who were found to have occult metastatic disease at surgical exploration, as compared to 15% of those who did not, with a diagnostic sensitivity and specificity of 66.7% and 81.6% when combined with conventional tumor markers [57]. Given their relative abundance across all stages, preoperative CHC elevations may therefore represent a more sensitive biomarker to judge the extent of disease burden and be less prone to sampling error, though it is important to note that data directly comparing CHCs to ctDNA are lacking. However, given the strengths and limitations of each of the aforementioned potential biomarkers, perhaps a multi-analyte panel that incorporates several blood-based biomarkers may ultimately provide the most comprehensive readout of occult metastatic risk.

Our findings additionally support the potential for CHCs to be utilized as a response biomarker in those receiving systemic therapies for PDAC, which is in line with and builds upon prior work evaluating patients with colorectal and esophageal malignancies [21]. We first found preoperative CHC levels to be significantly lower in patients who demonstrated complete or near-complete responses to NAT, which gives credence to the notion that these disseminated neoplastic cells may serve as functional readouts of tumor response. We then expanded this investigation to evaluate CHC levels prospectively and longitudinally from NAT initiation to completion, to surgical resection. While most patients saw decreased CHC levels upon treatment initiation, patients who demonstrated an ultimate partial pathologic response consistently maintained undetectable levels of CHCs while receiving treatment, whereas those with minimal responses saw CHCs persist through treatment. In both cases, CHC counts rose in the interval between NAT completion and surgical resection when no treatment was administered to levels consistent with elevations in CHC counts seen in the TRS 2/3 patients in our preoperative cohort of patients. Additionally, CHC levels rose in a single patient while receiving NAT prior to imaging evidence of the development of metastatic disease, thus further supporting their potential utility as functional readouts of evolving tumor biology.

CHC levels may be influenced by non-neoplastic inflammatory insults, which represents an important confounder to consider. There is precedent for this phenomenon, as prior work in mouse models has suggested that benign inflammatory insults promote colonic epithelial–immune cell fusion—an effect subsequently ameliorated with anti-inflammatory treatment [31,32]. Importantly, while cytokeratin is commonly used to identify circulating cancer cell populations (CHCs, CTCs, circulating cancer-associated macrophage-like cells [CAMLs]) [58,59,60], it itself is an epithelial marker used as a proxy for denoting epithelial-tumor identity and is not specifically a neoplastic marker. Given this, we uniquely assessed how systemic therapy may influence tumor-specific phenotypes of both tissue and CHCs in patients with metastatic PDAC. Broadly, disseminated CHCs appeared to mirror tumor tissue by proliferative index and, imperfectly, by several PDAC-specific markers. This suggests that CHCs may not only represent response biomarkers quantitatively but also qualitatively. While it would be impractical to serially perform core needle tissue biopsies for surveillance, which may only sample small sections of larger phenotypically heterogenous tumors, CHCs are logistically accessible and may be better templates upon which to assess response as they are inherently malignant cell populations with metastatic driving capacity that have selectively escaped tumors [20]. This work, while exploratory and in a limited sample size, is the first to suggest this relationship in human models and represents an important foundation upon which future work must expand on and validate.

Our study has several limitations to consider. First, given this was a single-institution study, we had a relatively small sample size, which limited our statistical power. Thus, we were unable to define CHC thresholds to predict the risk of metastasis or recurrence and perform prospective analyses using validation cohorts. Larger prospective studies are needed to validate these findings more rigorously. Additionally, the tumor specificity of circulating tumor cells (CHCs) may be limited, as these populations are identified using the epithelial marker CK and can be influenced by non-neoplastic inflammatory insults; however, this definition aligns with the characterization of other circulating neoplastic cell populations (CTCs, CAMLs). The identification of tumor-specific markers to distinguish neoplastic from non-neoplastic CHCs remains a critical area of future investigation and has the potential to enhance the tumor-specificity of CHCs as neoplastic biomarkers. Furthermore, while predetermined time points for longitudinal peripheral blood sample collections were established, the ultimate timing of collection of samples was influenced by patient availability and consent and was therefore not standardized across the entire cohort; however, multiple samples were collected per patient to ensure redundancy in the data. Lastly, as previously noted, core needle biopsies of tumor tissue represent imperfect measures of phenotypic response, as tissue specimens obtained may not be fully representative of overall tumor heterogeneity. However, these biopsies represent standard-of-care tissue sampling strategies, and clear phenotypic changes were seen in tissue samples across all four patients following treatment. Moreover, the WOO-M trial was designed for biomarker discovery following a short course (10 days) of treatment and was not designed for evaluation of ultimate survival characteristics; therefore, whether and how CHC phenotyping may inform prognostication remains an area of future investigation.

## 5. Conclusions

In conclusion, these data underscore the promising potential of CHCs to be utilized as blood-based biomarkers in the management of PDAC. Our findings indicate that CHCs are reliable predictors of disease burden and tumor response and have potential to be leveraged to personalize treatment strategies. These data present a compelling foundation for further exploration and validation of CHCs as biomarkers in larger, prospective, multicenter studies.

## Figures and Tables

**Figure 1 cancers-16-03650-f001:**
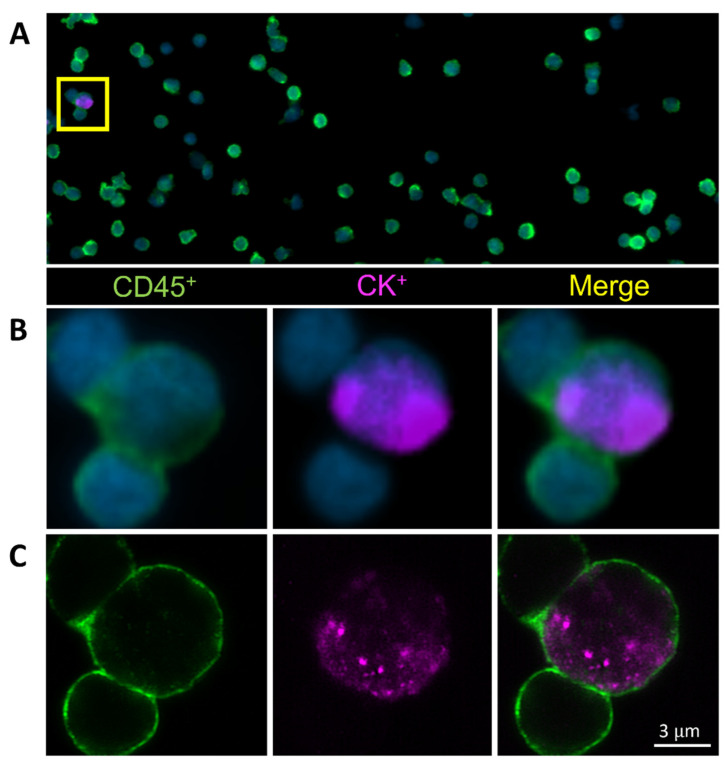
Circulating hybrid cells (CHCs) are detected in the peripheral blood of patients with pancreatic ductal adenocarcinoma (PDAC). Representative CHC was identified by co-expression of the pan-leukocyte protein CD45 (green) and the tumor epithelial protein pan-cytokeratin (CK, purple) utilizing (**A**,**B**) standard fluorescent microscopy and (**C**) confocal microscopy.

**Figure 2 cancers-16-03650-f002:**
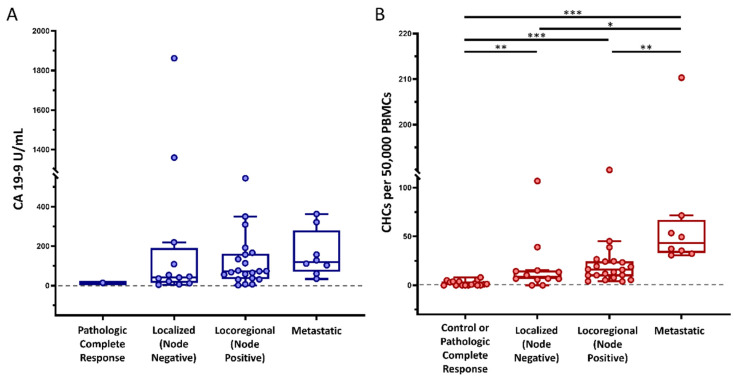
Circulating hybrid cell (CHC) levels across cancer stages. In our cohort of patients with pancreatic ductal adenocarcinoma (PDAC), (**A**) preoperative carbohydrate antigen 19-9 (CA 19-9) levels did not successfully discriminate between cancer stages, whereas (**B**) preoperative CHC counts increased with disease burden. *Peripheral blood mononuclear cells (PBMCs); adjusted p < 0.05 *, p < 0.01 **, p < 0.001 ****.

**Figure 3 cancers-16-03650-f003:**
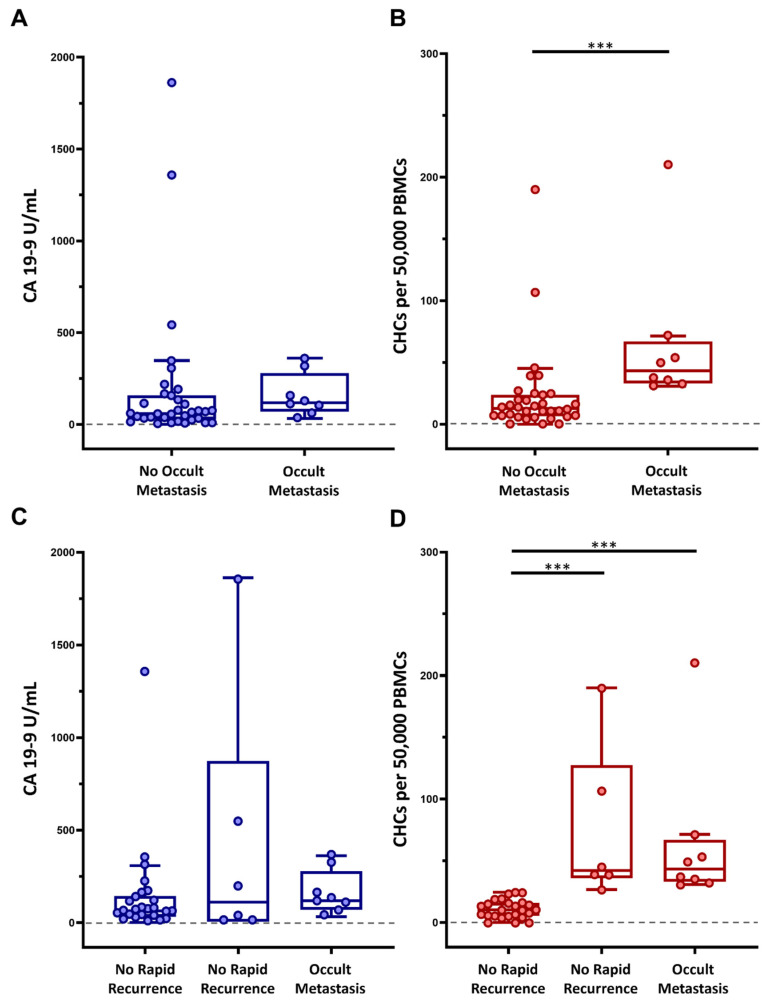
Preoperative circulating hybrid cell (CHC) levels as predictive biomarkers of occult disease. Performance of CA 19-9 and CHCs as a preoperative predictor of (**A**,**B**) occult metastatic disease and (**C**,**D**) rapid metastatic recurrence of disease within 6-months of margin-negative pancreatectomy, respectively. *Peripheral blood mononuclear cells (PBMCs); adjusted p < 0.001 ****.

**Figure 4 cancers-16-03650-f004:**
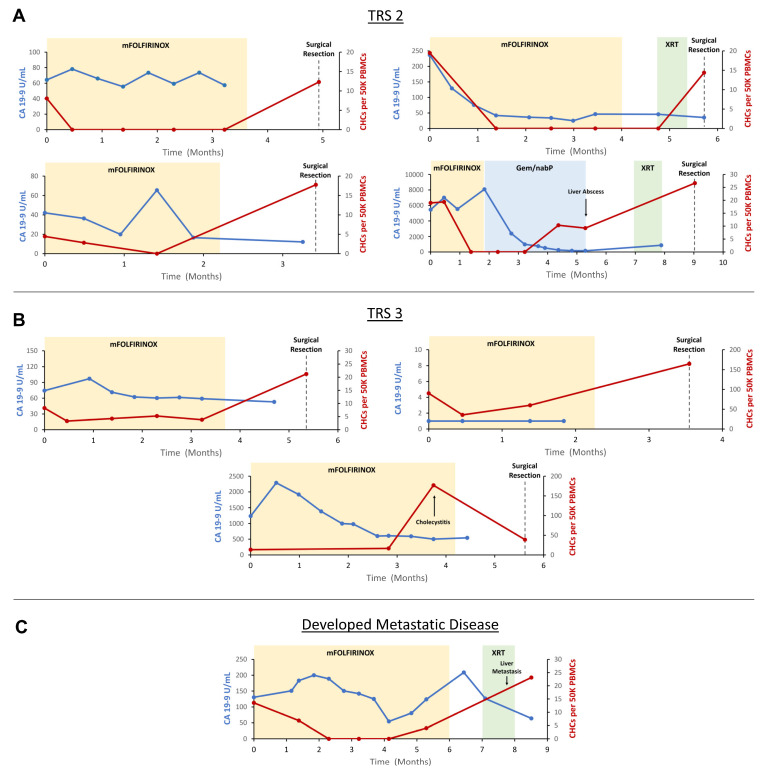
Longitudinal evaluation of circulating hybrid cells (CHCs) in patients treated with neoadjuvant therapy (NAT) for pancreatic ductal adenocarcinoma (PDAC). (**A**) n = 4 patients demonstrated partial pathologic response to NAT (TRS 2) and saw CHC counts drop undetectably low following initiation of NAT. (**B**) n = 3 patients demonstrated poor or no pathologic response (TRS 3) and CHCs persisted through treatment with NAT. (**C**) A single patient developed metastatic disease while being treated with NAT, and CHC levels rose prior to radiographic detection of metastases. CA 19-9 levels inconsistently correlated with response to NAT across the cohort. *Carbohydrate antigen 19-9 (CA 19-9); tumor response score (TRS); peripheral blood mononuclear cells (PBMCs); modified folinic acid, 5-fluorouracil, irinotecan, and oxaliplatin (mFOLFIRINOX); gemcitabine and nanoparticle albumin-bound paclitaxel (Gem/nab-P); chemoradiation (XRT)*.

**Figure 5 cancers-16-03650-f005:**
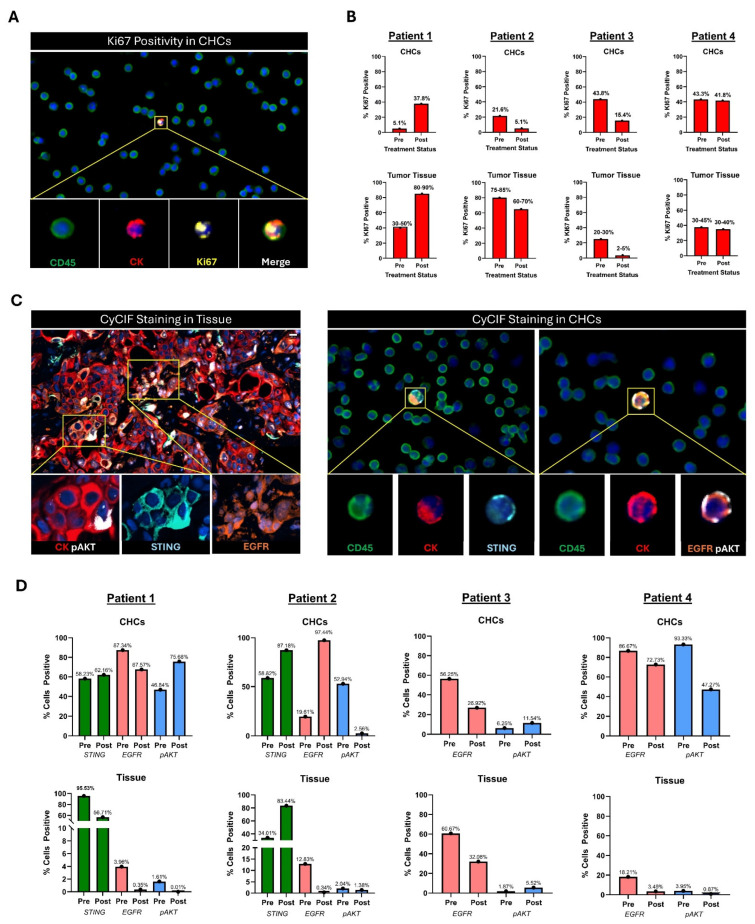
Analysis of paired metastatic tumor tissue and peripheral blood samples from four patients with metastatic PDAC. (**A**) Example of Ki67+ CHC. (**B**) Change in Ki67 expression between serial timepoints of CHCs compared to Ki67 expression in serial metastatic biopsies by a single expert pathologist. Biopsies and corresponding peripheral blood samples were collected before and after 10 days of treatment with systemic targeted therapy using either a MEK inhibitor (Patients 1 and 2 or a PLK-1 inhibitor (Patients 3 and 4). (**C**) Example of CyCIF staining in a metastatic tissue biopsy and CyCIF staining in peripheral blood slide showing expression of STING, EGFR, or pAKT. (**D**) Analysis of STING, EGFR, and pAKT expression across serial timepoints in CyCIF-stained peripheral blood slides and CyCIF-stained metastatic tumor biopsies. *Circulating hybrid cell (CHC), cyclic immunofluorescence analyses (CyCIF), epidermal growth factor receptor (EGFR), mitogen-activated protein kinase (MEK), phosphorylated protein kinase B (pAKT), polo-like kinase 1 (PLK-1), stimulator of interferon response cGAMP interactor (STING)*.

**Table 1 cancers-16-03650-t001:** Baseline Patient Demographic and Clinicopathologic Characteristics (n = 42).

Age at Diagnosis, Years, Median (IQR)	67 (59–74)
BMI, kg/m^2^, median (IQR)	25.98 (24.09–31.08)
Sex, no. (%)	
Female	18 (42.9)
Male	24 (57.1)
Race, no. (%)	
White	35 (83.3)
Asian	2 (4.8)
Pacific Islander	2 (4.8)
American Indian/Alaska Native	1 (2.4)
Unknown	2 (4.8)
Ethnicity, no. (%)	
Non-Hispanic	25 (83.3)
Hispanic	1 (2.4)
Unknown	6 (14.3)
Neoadjuvant therapy, no. (%)	
None	14 (33.3)
Chemotherapy only	24 (57.1)
Chemotherapy + XRT	4 (9.5)
Resectability status at diagnosis, no. (%)	
Resectable	23 (54.8)
Borderline resectable	13 (31.0)
Locally advanced	6 (14.3)
Pathologic stage, no. (%)	
0	1 (2.4)
IA	5 (11.9)
IB	4 (9.5)
IIA	3 (7.1)
IIB	13 (31.0)
III	8 (19.0)
IV	8 (19.0)
Liver	2 (25.0)
Peritoneal carcinomatosis	5 (62.5)
Retroperitoneal lymph nodes	1 (12.5)
Preoperative CA 19-9, U/mL, median (IQR)	69.45 (34.38–156.43)
No neoadjuvant therapy	105.15 (53.50–150.40)
Neoadjuvant therapy	63.45 (26.10–165.02)
Preoperative CHC count, per 50,000 PBMCs, median (IQR)	15.5 (7.1–34.6)
No neoadjuvant therapy	13.5 (10.2–21.5)
Neoadjuvant therapy	19.2 (6.8–40.4)

Abbreviations: IQR, interquartile range; BMI, body mass index; CA 19-9, carbohydrate antigen 19-9; CHC, circulating hybrid cell; no., number; PBMC, peripheral blood mononuclear cells; XRT, chemoradiation.

## Data Availability

The datasets supporting the conclusions of this article are included within the article (and its additional files) and available from the corresponding author on reasonable request.

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
