# Peer review of "Circulating Neoplastic-Immune Hybrid Cells Are Biomarkers of Occult Metastasis and Treatment Response in Pancreatic Cancer"

_cancers, 2024, doi:10.3390/cancers16213650_

Round 1

Reviewer 1 Report

Comments and Suggestions for Authors

I would like to put forward a suggestion to the authors and specifically to expand the Discussion section by addressing the limitation of the study, with particular emphasis on the specificity of markers used for the identification of hybrid cells in comparison to other circulating marker such as circulating tumour cells.

In addition, it would be valuable to include information on the prevalence of these cells in healthy controls. The authors mention that an inflammatory state could induce the appearance of these cells, which represents a significant limitation in the study.

Reviewer 2 Report

Comments and Suggestions for Authors

This is a very interesting study, and the study may have significant application during PDAC patient clinical trial and treatment. Easy-to-understanding methods with sufficient details were provided, which makes the result section deliver more confidence to this reviewer. After following the suggestions/comments below, this manuscript should be ready for publication in Cancers.

Specific comments are below.

1. The reference 1 may not be the most appropriate and authority publication to be cited here. The link provided below is the more appropriate and authority publications to be cited. Both publication-provided data analysis and/or figures indicated the same conclusion that PDAC is predicted to become the second leading cause of cancer-related deaths 56 in the United States by 2030.

https://www.ncbi.nlm.nih.gov/pubmed/24840647

https://jamanetwork.com/journals/jamanetworkopen/fullarticle/2778204

2. The definition of circulating hybrid cells (CHCs) or circulating neoplastic-immune hybrid cells (CHCs) should be consistent throughout the manuscript. Or this will confuse readers.  All others (if any) should be the same to be consistent throughout the MS as a general rule.

3. These authors wrote “On each slide, one region was left unstained to be utilized as a reference control for detecting autofluorescence during analysis”. Question: how these authors could easily control one region on each slide left unstained as a reference control. Please explain.

4. Can these authors explain in more detail about “For each peripheral blood sample, at least 50,000 cells were enumerated, and CHC numbers were normalized to 50,000 total PBMCs.”? What does the normalized to 50K total PBMCs mean?

5. The Method Section 2.5 for Ab-oligo staining following by steps including adding distinct fluorophores should be briefly explained why this is needed before distinct conjugated fluorescent antibodies. This will be very good for some readers who may not be familiar with the technology. Please explain it for me. Additionally, why the Ab-oligo needs to be used in the studies? This may need to be explained in the Result section and/or in the Discussion section. Please explain these to me and relevant stuff can be used in the revised manuscript.

6. Figure 1 should indicate the CD45 and CK represent immune cell or tumor cells again for easy catch up by readers. Similar thing for other figures if existed. Remember, facilitate audience reading will be a good thing to let them like your research.

7. When possible (space available), the font in figures should be large enough for readability. Figures 2, 3, 4, 5 fonts in most places/cases are too small.

8. Figure 5ABCD and patients ABCD were used the same format font in the figure legend description. This is easy to be confused. Why not use Patients 1, 2, 3, 4 instead of using ABCD in figure and legend for easier description, which will be much clear without confusing.

9. Please unify the patients mentioned in Figure 4 and Figure 5. These authors should clearly indicate whether the same patients with different analysis or different patients with different analysis. If it is the latter, why not use the same patients but used different patients for a justification, which will need to be explained in somewhere and may be discussed in the Discussion section.

Comments on the Quality of English Language

No big issue.
